# Injection Near the Stratopause Mitigates the Stratospheric Side Effects of Sulfur-Based Climate Intervention

Pengfei Yu[1]*, Yifeng Peng[2], Karen H. Rosenlof[3], Ru-Shan Gao[3,4], Robert W. Portmann[3], Martin Ross[5], Eric Ray[3,4], Jianchun Bian[6], Simone Tilmes[7] and Owen B. Toon[8]

[1] Institute for Environmental and Climate Research, College of Environment and Climate, Jinan University, Guangzhou, China

[2] School of Atmospheric Sciences, Lanzhou University, Lanzhou, China

[3] Chemical Science Laboratory, National Oceanic and Atmospheric Administration, Boulder, Colorado

[4] Cooperative Institute for Research in Environmental Sciences, University of Colorado Boulder, Boulder, CO, USA

[5] Civil and Commercial Launch Projects, The Aerospace Corporation, Los Angeles, California

[6] Key Laboratory of Middle Atmosphere and Global Environment Observation, Institute of Atmospheric Physics, Chinese Academy of Sciences, Beijing, China

[7] Atmospheric Chemistry, Observations, and Modeling Laboratory, National Center for Atmospheric Research, Boulder, Colorado

[8] Department of Atmospheric and Oceanic Sciences and Laboratory for Atmospheric and Space Physics, University of Colorado, Boulder, Colorado

*Correspondence to*: Pengfei Yu (pengfei.yu@colorado.edu)

**Abstract.** Stratospheric aerosol injection (SAI) using sulfur has been proposed to cool the planet by reflecting sunlight 20 back to space. A commonly proposed SAI, with sulfur dioxide injection rate of 10 Tg/year at 25 km, accumulates aerosols in the tropical lower stratosphere, causing a 6 K warming of the tropical lower stratosphere that impact the entry value of stratospheric water vapor and jet positions. This approach could also delay October Antarctic total column ozone (TCO) recovery to 1980s values by 25-55 years. We propose a novel SAI approach of injecting sulfur at 50 km (SAI$_{50}$) that substantially reduces these negative impacts. In SAI$_{50}$, the mean meridional overturning circulation near the stratopause 25 rapidly transports aerosols to mid-high latitudes, preventing their accumulation in the tropical lower stratosphere. This approach reduces tropical stratospheric warming to 3 K and shortens the Antarctic ozone recovery delay to 5 years. Furthermore, SAI$_{50}$ demonstrates greater cooling efficiency, enhancing global and polar surface cooling by 22% and 40% respectively. Consequently, SAI$_{50}$ preserves 20% more Arctic September sea ice compared to lower-altitude SAI. These findings suggest that SAI$_{50}$ could offer a more effective and less disruptive approach to climate intervention.

# 1 Introduction

Stratospheric aerosol injection (SAI) has been proposed to counteract global warming at the surface by injecting sulfur into the stratosphere where it reflects a portion of sunlight back into space. Climate model simulations of SAI (Tilmes et al., 2021; Tilmes et al., 2018b) have demonstrated its potential to keep the Earth from warming beyond a certain threshold, such as 1.5 or 2.0 degrees Celsius above preindustrial temperatures, and to preserve the sea ice (Lee et al., 2023b).

The deployment of SAI may result in warming of the lower stratosphere, increases in stratospheric water vapor, reductions in precipitation and increases in acid-rain deposition relative to the climate without SAI (Macmartin et al., 2022; Tilmes et al., 2018b). A major climate concern associated with SAI is tropical stratospheric heating and the resultant changes in the tropospheric weather and climate (Ferraro et al., 2015; Visioni et al., 2021; Wunderlin et al., 2024). Previous studies (Tilmes et al., 2021; Tilmes et al., 2018b) show that SAI sulfate aerosols tend to accumulate in the tropical lower stratosphere, where the sulfur is injected. This accumulation of sulfate aerosols leads to significant warming of the tropical cold-point tropopause and the lower stratosphere, as the sulfate aerosols absorb both upward long-wave radiation and downward solar radiation. A warmer tropical tropopause allows more water vapor to enter the stratosphere, resulting in further surface warming. This partly offsets the cooling effect intended by SAI (Tilmes et al., 2018a; Visioni et al., 2021). Furthermore, the tropical lower stratospheric warming caused by SAI strengthens the polar jets (Ferraro et al., 2015; Tilmes et al., 2009; Visioni et al., 2020). It also weakens the subtropical jets (Ferraro et al., 2015; Tilmes et al., 2018a), subsequently shifting weather patterns including the precipitation belts, dry zones and storm tracks.

The delay in the recovery of Antarctic ozone is another risk associated with SAI deployment. SAI modeling studies (Macmartin et al., 2022; Tilmes et al., 2020; Tilmes et al., 2021; Tilmes et al., 2018b) have typically simulated sulfur injections at 20 and 25 km in the tropics and midlatitudes. The sulfate particles from SAI are transported to the polar lower stratosphere through the Brewer-Dobson Circulation (BDC). In the absence of SAI and under high greenhouse gases emission scenarios (RCP 8.5), it is projected that the October Antarctic total column ozone (TCO) could return to 1980s level of ~290 Dobson Units (DU) between the 2040s and 2050s (Tilmes et al., 2021). However, when SAI is deployed to achieve the 1.5°C temperature goal under the RCP8.5 emission scenario, the simulated ozone recovery is delayed by 25 to 55 years (Tilmes et al., 2021). Similarly, to meet the same temperature goal in a moderate emission scenario of SSP2-4.5, a 20-year delay in ozone recovery is estimated with SAI deployment starting from year 2035 (Macmartin et al., 2022).

Tropical injection leverages the ascending branch of the BDC to efficiently transport aerosols into the global stratosphere, producing cooling across hemispheres and more effective surface cooling, while equatorial injection leads to substantial overcooling in the tropics and residual surface warming in the high latitudes (Kravitz et al., 2019; Tilmes et al., 2018b). High-latitude injections reduce the stratospheric warming, enhance polar cooling and sea ice preservation compared with tropical injection strategies (Lee et al., 2021; Lee et al., 2023b). However, it requires larger injection amounts to

achieve the same global cooling as tropical injections due to the shorter aerosol lifetime (Henry et al., 2024; Zhang et al., 2024).

In this study, we propose a novel SAI approach which substantially reduces both stratospheric warming and Antarctic ozone loss. Our new approach involves injecting $SO_2$ near the stratopause at 50 km ($SAI_{50}$) instead of the lower altitudes used in previous SAI studies (Macmartin et al., 2022; Tilmes et al., 2020; Tilmes et al., 2021; Tilmes et al., 2018b). The enhanced depletion of Antarctic ozone is inevitable when implementing sulfur-based SAI approaches. To minimize the Antarctic ozone loss, it is essential that some sulfate aerosols from the intervention remain at high altitudes in the polar stratosphere. By doing so, high-altitude sulfate aerosols reduce $NO_x$ levels, slowing $NO_x$-driven ozone loss and allowing ozone to accumulate in the middle stratosphere, which can offset the ozone loss caused by reactive halogen species in the lower stratosphere. In addition, aerosols formed at higher altitudes are rapidly transported to the mid-high latitudes rather than accumulating in the tropical lower stratosphere. This significantly reduces the tropical stratospheric heating when compared to lower-altitude SAI approaches. In addition, higher AOD in mid-high latitudes with $SAI_{50}$ produces more surface cooling compared with $SAI_{25}$ and consequently helps to preserve the sea ice at both Poles.

## 2 Materials and Methods

### 2.1 Global climate model: WACCM-MAM3

We use the Community Earth System Model, version 1 (CESM1-WACCM) (Hurrell et al., 2013) coupled with the three-mode version of the Modal Aerosol Model (MAM3) (Liu et al., 2012) to study the climate response to SAI. MAM3 provides a physically-based treatment of aerosol size, mixing, and key microphysical processes, including nucleation, growth, deposition, and interactions with clouds and precipitation (Liu et al., 2012). The nucleation of sulfate aerosol is produced from aqueous-phase $SO_2$ oxidation and to a lesser extent from $H_2SO_4$ condensation on pre-existing aerosol (Liu et al., 2012). The horizontal resolution of the model is 1.89 degrees latitude by 2.5 degrees longitude, with 70 vertical levels ranging from surface to 145 kilometers. The model calculates size-dependent aerosol optical properties at each model vertical level, accounting for local temperature and relative humidity conditions that vary with altitude. The Rapid Radiative Transfer Model for Global Climate Models (RRTMG) with a two-stream algorithm for multiple scattering is used to simulate the radiative feedback (Iacono et al., 2008). The radiative effects of $SO_2$ are not included in the model, as these effects are negligible for Pinatubo-scale $SO_2$ injections (Osipov et al., 2020).

A full chemistry module, including both gas-phase and heterogeneous chemistry, is coupled with MAM3 (Kinnison et al., 2007). This chemical scheme includes 74 photochemical reactions, 151 gas-phase chemical reactions and 17 heterogeneous chemical reactions (Emmons et al., 2010). Note that the photolysis of $H_2SO_4$ gas (Mills et al., 2005; Vaida et al., 2003) is not included in the model.

## 2.2 SAI Injection scenarios

In SAI experiments with injection height of 20, 25, 35 and 50 km, $SO_2$ was continuously injected at two model grid boxes located at (15°N, 0°E) and (15°S, 0°E) with a total rate of 10 Tg per year (i.e. 5 Tg in each grid box). To study the Antarctic ozone response to SAI under different Ozone Depleting Substance, ODS, concentrations, the model is run with ODS concentrations fixed in year 2000, 2040 and 2065, respectively. All the simulations include 3 ensemble members, and each member is performed for 15 years with an additional 5-year model spin-up. For simulations of year 2000, model is initialized with atmospheric ODS and Greenhouse Gases (GHGs) conditions of year 2000. For simulations of year 2040 (2065), the ODS and GHGs are fixed in the year of 2040 (2065). To compare the surface temperature and sea ice extent responses to two SAI approaches, 50-km SAI ($SAI_{50}$) and 25-km SAI ($SAI_{25}$), the model is run with coupled ocean, land, atmosphere and sea ice components for 45 years with an additional 10-year model spin-up for each SAI approach and the control simulation without SAI.

We chose $SO_2$ as the SAI injection material because it is the primary sulfur compound in volcanic eruptions, providing natural analogues for model validation. While $H_2S$ gas could offer significant mass efficiency advantages (approximately 2x reduction in required lifting mass compared to $SO_2$), its toxicity concerns and climate risks warrants future investigation.

## 3 Results

### 3.1 Model validation

We use the Whole Atmosphere Community Climate Model (WACCM) to study the spatial distributions of sulfate aerosols from $SAI_{50}$ and their impacts on stratospheric heating, surface cooling, sea ice preservation and ozone chemistry. The details of the model are provided in the Materials and Methods. The model used reproduces the observed stratospheric aerosol perturbation following the 1991 Pinatubo volcanic eruption (Mills et al., 2016) and the 2022 Hunga Tonga-Hunga Ha'apai volcanic eruption with the observed plume reaching the stratopause (Proud et al., 2022; Zhu et al., 2022). Shown in Fig. 1a, we compare the simulated temporal evolution of the global mean aerosol optical depth (AOD) anomalies following the eruptions to observations by the Advanced Very High-Resolution Radiometer (AVHRR) (Zhao et al., 2013) and the Global Space-based Stratospheric Aerosol Climatology (GloSSAC) (Kovilakam et al., 2020). The model results are similar for the observed peak and the decay rate of stratospheric AOD anomalies, demonstrating the model can reasonably simulate the transport and chemistry of aerosols from large and high-altitude volcanic eruptions, which are natural analogues of SAI. The spread across our simulations of 45 ensemble members can capture the natural variations in stratospheric circulation. Comparison with MERRA2 reanalysis data (2000-2020) shows reasonable agreement in key stratospheric metrics including temperature at 100 hPa, Quasi-biennial Oscillation (QBO) strength, Semiannual Oscillation (SAO) strength (1 hPa tropical zonal winds), and polar vortex strength (Fig. S1), providing confidence that our simulations represent a realistic range of

possible stratospheric conditions. While the model has known biases in polar processes (Ern et al., 2018; Garcia et al., 2017), it captures the fundamental features of stratospheric circulation relevant for simulating SAI aerosol transport and distribution.

**3.2 High-altitude SAI at 50 km**

Instead of a pulse injection as would be the case for a volcanic eruption, in our SAI experiment we inject $SO_2$
continuously at 15°N and 15°S at a rate of 10 Tg per year. The amount of $SO_2$ is on the order of what prior studies injecting at 20 km required for maintaining 2020 surface temperatures in the year 2030 under the RCP8.5 emission scenario. The modeled global mean sulfate AOD increases with time and reaches a plateau three years after initial injection. The simulated AOD anomaly with a tropical injection at 25 km is about 30-50% larger than when injecting at 20 km, consistent with previous studies (Tilmes et al., 2018b; Lee et al., 2023a). Our study further shows that the simulated AOD with injections at
35 km is approximately 10% larger than for a 25 km-injection (Fig. 1a). The simulated global mean AOD of $SAI_{50}$ doesn't show further enhancement compared to that of 35-km injection, although the lifetime of total sulfur (in both gaseous and condensed phases) is 13% longer (Fig. S2). With 50-km injection, a higher fraction of sulfur resides in the gaseous phase in the equilibrium state due to the relatively high vapor pressure of sulfuric acid at the ambient temperature (Fig. S2). It's important to note that while $SO_2$ is injected at 50 km, the actual sulfate aerosol formation occurs at much lower altitudes
(primarily between 10-30 km) due to the rapid transport of precursor gases and more favorable conditions for aerosol formation at lower altitudes. Above 40 km, the simulated stratospheric sulfur species primarily exist in the form of $SO_2$, with ~3 orders of magnitudes higher than $H_2SO_4$ (Fig. S3).

The latitudinal distributions of AOD are similar for all lower altitude injections (at 20 km, 25 km and 35 km), with higher AOD simulated in the tropics and the mid-latitudes (Fig. 1b). However, for $SAI_{50}$, the simulated AOD is weighted
towards higher latitudes, with smaller AOD anomalies found in the tropics. In polar regions, the simulated AOD is a factor of 2-3 higher than that for $SAI_{25}$. In the lower-altitude injection scenarios, the injected sulfur species are primarily transported upwards at low latitudes following the upward branch of BDC; while for $SAI_{50}$, the sulfur species are rapidly transported latitudinally via mean meridional circulation near the stratopause.

As shown in Fig. 1c, the simulated annual mean aerosol surface area density (SAD) over the Antarctic peaks around 12
145 km and decreases with altitude for the 20-km injection scenario. However, for higher altitude injection scenarios (at 25, 35 and 50 km), there is an additional aerosol peak simulated in the middle stratosphere around 25 km. Note that the sulfate aerosol evaporates into sulfuric acid gas above 35-40 km but reforms when the gas is transported to lower altitudes (10-30 km) via large-scale circulation. The simulated SAD in the Antarctic stratosphere increases significantly with the injection height as more aerosols accumulate at the poles with higher injection height. At 25 km altitude, the simulated annual mean
SAD with $SAI_{50}$ is about ten times higher than that of $SAI_{25}$. In the lower stratosphere at ~12 km, the simulated SAD with $SAI_{50}$ shows a weaker enhancement of about 20% compared to $SAI_{25}$. In summary, with the same injected amount, 10% more global mean AOD is simulated with $SAI_{50}$ compared to $SAI_{25}$. In addition, more aerosols are transported into the middle stratosphere at high latitudes with $SAI_{50}$ relative to the lower altitude injection scenarios. The distinct latitudinal and

vertical distributions of aerosols in SAI$_{50}$ are expected to influence the climate cooling benefits and mitigate the associated
stratospheric impacts, as detailed in the following subsections.

### 3.3 Reduced tropical lower stratospheric warming

One climate concern of SAI is that the injected sulfate aerosol accumulates in the tropical lower stratosphere, absorbing upward longwave and downward shortwave radiation, and warming the tropopause. This stratospheric warming produces undesired climate consequences including enhanced stratospheric water vapor, strengthening of the polar jets and weakening of the subtropical jets. As shown in Fig. 2a-b, SAI$_{25}$ with an injection rate of 10 Tg per year warms the tropical lower stratosphere, reaching an annual mean warming of 6 K at the tropopause. Raising the injection altitude to 50 km (SAI$_{50}$) reduces this tropical tropopause warming to 3 K. The magnitude of warming in the lower stratosphere in the tropics is significantly reduced with SAI$_{50}$ compared to SAI$_{25}$. This is primarily due to the rapid transport of aerosols in SAI$_{50}$ into mid-high latitudes rather than accumulating in the tropical lower stratosphere.

When the tropical tropopause is warmer, more water vapor enters the stratosphere, perturbing the stratospheric chemical and radiative budget (Tilmes et al., 2018a; Visioni et al., 2021). The simulated stratospheric mean water vapor mass mixing ratio anomalies are about 0.5-0.8 and 0.1-0.3 parts per million for SAI$_{25}$ and SAI$_{50}$, respectively (Fig. 2c-d). This corresponds to an increase in the stratospheric water vapor burden by 340 Tg (or ~15%) and 90 Tg (or ~4%) for SAI$_{25}$ and SAI$_{50}$, respectively. There is a positive feedback when stratospheric water increases near the tropopause (Dessler et al., 2013; Tilmes et al., 2018a), this will be weaker for the SAI$_{50}$ case. In addition to the water vapor increase, the subtropical jets are weakened, while the polar jets are strengthened in response to the tropical stratospheric warming (Fig. 2e), which is consistent with previous work (Lorenz and Deweaver, 2007; Woollings et al., 2023). However, changes in the strength of the subtropical and polar jets are significantly reduced with SAI$_{50}$ (Fig. 2f). While these jet stream changes typically influence precipitation patterns and storm tracks (Lu et al., 2007; Mbengue and Schneider, 2013), detailed analysis of precipitation responses lies outside the scope of this study, as regional precipitation changes carry larger uncertainties in climate models.

### 3.4 Enhanced global mean and polar surface cooling and sea ice preservation

Due to the mean meridional overturning circulation near the stratopause, the injected sulfur is transported from the injection latitude to the winter pole and then downwards. As a result, the aerosols are mostly distributed in the altitude range of 10-30 km in the mid-high latitudes. In the mid-high latitudes, the larger AOD simulated for SAI$_{50}$ (as shown in Fig. 1b) leads to more surface cooling compared to SAI$_{25}$. Shown in Fig. 3a, SAI$_{50}$ induces extra surface cooling by 1-3 K at both poles (60-90°N, 60-90°S) compared to SAI$_{25}$, while maintaining similar cooling effects in the tropics. SAI$_{50}$ exhibits a 22% greater global mean surface cooling and a 40% greater polar surface cooling compared to SAI$_{25}$ (Fig. 3b). The warmer temperatures in the North Atlantic under high-altitude injection (Fig. 3a) reflect differences in ocean circulation response. While greenhouse gas forcing typically weakens the Atlantic Meridional Overturning Circulation (AMOC), creating a

185 characteristic 'warming hole', the more effective cooling from $SAI_{50}$ partially offsets this AMOC weakening. This maintained ocean heat transport appears as a relative warming signal in the North Atlantic region.

In SAI50, the simulated 22% greater global mean surface cooling compared to the 10% increase in global mean AOD (Fig. 1a) primarily reflects the higher proportion of aerosols distributed at high latitudes, where Arctic amplification mechanisms enhance the cooling efficiency of aerosol forcing. Arctic amplification processes, including ice-albedo feedback

and stable atmospheric conditions (Barnes and Polvani, 2015), contribute to this enhanced regional cooling response. A minor contribution also comes from the reduced stratospheric water vapor enrichment (Fig. 2c-d). The zonal temperature response shows significantly stronger cooling in $SAI_{50}$ compared to $SAI_{25}$, particularly in middle and high latitudes, where the differences exceed the internal variability of either scenario (Fig. S4). Fig. 3c and Fig. 3d compare the simulated seasonal cycles of the surface temperature anomalies between $SAI_{50}$ and $SAI_{25}$ at each pole, respectively. In the Arctic (60-90°N),

$SAI_{50}$ induces a surface cooling of about 7-8 K in the fall and 3-4 K in the spring relative to the simulations without SAI. This cooling effect is 1-2 K more pronounced than that achieved by $SAI_{25}$ throughout the entire year. In the Antarctic (60-90°S), the simulated surface cooling remains at about 1 K for $SAI_{25}$, while for $SAI_{50}$, it ranges from 1.5 to 2.5 K. The Arctic cooling exhibits more pronounced seasonality, with maximum effects during fall-winter seasons (Fig. 3c). This seasonal pattern aligns with the mechanism of Arctic amplification, which is driven by increased outgoing longwave radiation and

heat fluxes from areas of seasonal sea ice loss during October-April (Dai et al., 2019). In contrast, Antarctica's year-round ice cover results in more uniform cooling throughout the year (Fig. 3d).

The enhanced cooling effect in high latitudes should result in less sea ice loss (Lee et al., 2023b). When employing $SAI_{25}$, the simulated Arctic sea ice extent in September (i.e. the minimum extent month) expands to 36% of the peak value simulated in March (i.e. the maximum extent month), as depicted in Fig. 3c. However, by using the same amount of injection

but at a higher injection altitude ($SAI_{50}$), the simulated Arctic September sea ice extent further increases to 56% of the March extent, resulting in the preservation of 20% more sea ice in the Arctic September compared to $SAI_{25}$. In the Arctic, $SAI_{50}$ preserves more sea ice than $SAI_{25}$ during the summer and fall (from June to December), while maintaining a similar sea ice extent during the winter and spring compared to $SAI_{25}$. In contrast, the simulated sea ice extent in the Antarctic is elevated by 1-2 million $km^2$ throughout the entire year with $SAI_{50}$, representing about a 100% increase relative to that achieved with

$SAI_{25}$ (Fig. 3d). While our 45-year ensemble simulations cannot capture century-scale deep ocean circulation adjustments, they do capture the primary thermodynamic response of sea ice response to SAI: the direct radiative cooling effect, sea ice response to this cooling, upper ocean adjustments, and ice-albedo feedback.

WACCM model has demonstrated strong performance in reproducing observed post-Pinatubo responses, including the GMST cooling of approximately 0.5°C (Solomon et al., 2011). The simulated GMST anomalies in both $SAI_{25}$ and $SAI_{50}$

scenario significantly exceed the natural variability (one standard deviation) of 0.25 K calculated from ensemble members (Fig. S4), suggesting a robust response to the intervention. The global mean precipitation reduction in $SAI_{50}$ is approximately 20% greater than in $SAI_{25}$, proportional to its stronger cooling effect, with both signals also exceeding natural variability (Fig. S4). While CESM-WACCM has demonstrated skill in simulating large-scale tropospheric circulation patterns (Peings et al.,

2017; Simpson et al., 2020), we acknowledge that detailed tropospheric responses, particularly regional precipitation and deep ocean circulation changes, carry larger uncertainties - a limitation common to current climate models. Future work using multi-model ensembles would be valuable for better constraining these tropospheric responses. Additionally, while this study uses idealized fixed-rate injections to compare fundamental differences between injection heights, more practical implementation would require varying injection rates to meet specific climate objectives (Henry et al., 2024; Macmartin et al., 2022; Tilmes et al., 2018b). The enhanced high-latitude cooling observed in $SAI_{50}$ suggests potential advantages for offsetting Arctic amplification, though determining optimal injection strategies would depend on defined climate goals and metrics.

### 3.5 Reduced Antarctic ozone depletion

Aerosols play a significant role in influencing stratospheric ozone concentrations through changes in transport, as well as through both homogeneous and heterogeneous chemical reactions (Solomon, 1999). The ozone chemical response to elevated aerosol SAD is sensitive to altitude and season. In the lower stratosphere, sulfate aerosols provide surfaces for various heterogeneous reactions, releasing reactive Cl and Br radicals that contribute to ozone depletion. In the middle stratosphere, sulfate aerosols facilitate the hydrolysis of $N_2O_5$, leading to the depletion of $NO_x$ and consequently slowing down the ozone destruction via the $NO_x$ catalytic cycle. In addition to the chemical processes, total column ozone (TCO) is also affected by dynamical processes, including eddy transport and changes in strength of the Brewer-Dobson circulation (BDC). Shown in Fig. S5a-b, simulated ozone anomalies are negative in the lower stratosphere and positive in the middle stratosphere. The simulated positive ozone anomalies in the middle stratosphere due to $NO_x$ depletion are more significant in both hemispheres using $SAI_{50}$, because more aerosols are distributed in there compared to $SAI_{25}$. The simulated October Antarctic TCO shows ~12% depletion with $SAI_{25}$, while only about 4% depletion with $SAI_{50}$ when injecting 10 Tg $SO_2$ each year under the Ozone Depleting Substances, ODS, condition of the year 2040, following the RCP8.5 emission scenario (Fig. S5c). The reduced depletion of Antarctic TCO with $SAI_{50}$ is attributed to the higher sulfate concentration simulated in the middle stratosphere, which leads to the depletion of $NO_x$.

The Antarctic ozone depletion is closely related to the atmospheric ODS concentrations (Macmartin et al., 2022; Solomon et al., 2016), which have decreased significantly since the global commitment to the Montreal Protocol and subsequent treaties. As illustrated in Fig. 4a, the mole mixing ratio of CFC-11, one of the most important ODS, is expected to decline from ~270 parts per trillion (ppt) in the year 2000 to around 80 ppt between the 2060s (SSP5-8.5) and the 2080s (SSP2-4.5). As a result of the decline in ODS, simulated Antarctic TCO in October is expected to increase from 200 to ~320 DU in early 2060s with SSP5-8.5 (or early 2080s with SSP2-4.5). According to the previous geoengineering simulations (Macmartin et al., 2022; Tilmes et al., 2021), in order to prevent surface temperatures from exceeding 1.5 °C relative to preindustrial levels based on moderate and high emission scenarios, we would need to inject 3-12 Tg of $SO_2$ per year in the 2040s and 10-20 Tg of $SO_2$ per year in the 2060s. Figure 4a shows that the simulated maximum depletion of the October

Antarctic TCO due to SAI with 25-km injection of 10 Tg/year would be around 30 DU if SAI is implemented from the 2040s to the late 21$^{st}$ century.

The simulated October Antarctic TCO reduction due to SAI$_{50}$ is about one third of the depletion simulated with SAI$_{25}$ (Fig. 4a). Notably, in the 2060s (RCP8.5) or 2080s (SSP4.5), the simulated TCO with SAI$_{50}$ shows no reduction relative to the case without SAI. As shown in Fig. 4b-c, SAI$_{50}$ leads to less column ozone reduction both below and above 18 km in the Antarctic, with a more significant difference above 18 km compared to SAI$_{25}$. The higher SAD above 18 km over the Antarctic, as simulated in the SAI$_{50}$ scenario (Fig. 1c), results in more depletion of NO$_x$ via N$_2$O$_5$ hydrolysis. Consequently, less ozone is lost through the NO$_x$-catalytic reactions, contributing to more ozone in the middle stratosphere and reduced TCO depletion simulated with SAI$_{50}$. Our findings demonstrate that SAI$_{50}$ is more efficient in preserving the Antarctic TCO compared to SAI$_{25}$, due to the differences in aerosol distribution and subsequent chemical reactions. The delay in recovery to 1980 Antarctic TCO levels is about 5 years with SAI$_{50}$. While placing aerosols at higher altitudes (50 km) could potentially increase surface UV-B radiation due to photons being scattered through shorter optical paths in the ozone layer (Madronich et al., 2018), our simulations show enhanced ozone concentrations in the SAI$_{50}$ scenario compared to SAI$_{25}$. The net effect on surface UV radiation would depend on the balance between these competing processes and requires further detailed evaluation.

## 4 Summary and Outlook

Figure 5 presents a summary of climate impacts of interest resulting from SAI using two different injection scenarios: a 25-km injection (SAI$_{25}$) and a 50-km injection (SAI$_{50}$). SAI$_{50}$ appears to result in some reduced adverse climate impacts relative to those caused by SAI$_{25}$. These benefits include reduced tropical stratospheric warming, which in turn reduces changes in the stratospheric water budget and the strength of subtropical and polar jets. Additionally, SAI$_{50}$ results in a 67% reduced depletion of Antarctic TCO, with the delay in Antarctic ozone recovery being shortened from the previously reported 25-55 years to about 5 years. Furthermore, SAI$_{50}$ offers enhanced climate benefits compared to SAI$_{25}$ with the same amount of injection, such as a 10% higher global mean AOD, with a 22% greater global mean surface cooling. SAI$_{50}$ also provides 40% more surface cooling in the polar region, resulting in a 20% greater preservation of Arctic sea ice in September. While our results demonstrate reduced stratospheric risks and enhanced polar cooling with high-altitude injection, fundamental challenges common to all SAI approaches remain. These include termination shock, multi-decade deployment commitment, and potential long-term impacts on ocean circulation that could modify the polar temperature and sea ice response patterns through altered poleward heat transport and vertical mixing processes. Future studies with multi-century simulations are needed to fully capture deep ocean adjustment and its effects on the climate response patterns identified here.

Conventional aviation technology limits jet engine-based SAI injection applications to less than 20 km altitude. Design studies suggest exotic propulsion systems could achieve a further 5 km in altitude (Lockley et al., 2020) but still far below the requirements of high-altitude SAI (SAI$_{50}$). Upper stratospheric SAI injection could be done with a fleet of reusable

rockets, flying suborbital trajectories, with engines using $H_2/O_2$ propellants. Hydrogen fueled rocket engines emit mainly $H_2O$ which would have a negligible impact on climate and ozone, even at very high emission rates (Larson et al., 2017). The benign emissions from a hydrogen fueled rocket platform contrasts with the complex BC, $CO_2$, and NO emissions from a kerosene fueled engine, rocket or jet, so that hydrogen would be the preferred fuel for SAI propulsion.

Based on SSP2-4.5 scenario, achieving the 1.5-degree temperature goal would require an annual $SO_2$ injection rate of 3-8 Tg/year during 2040-2060 (Macmartin et al., 2022). Delivering 3-8 Tg of $SO_2$ per year to 50 km altitude could be done with a fleet of 30-80 reusable rockets each with a 500-ton payload, and each launched every other day. Though detailed engineering analysis of a 50 km SAI injection suborbital launch system has not yet been done, the concept is well within the scope of current technology (Chang and Chern, 2021; Larson et al., 2017) and recent spaceflight experience. Indeed, the requirements of a $SAI_{50}$ rocket-based injection system overlap with requirements and goals of other technologies such as rapid point-to-point rocket cargo that require low-cost routine operations (Chang and Chern, 2021). Our results clearly indicate that a detailed engineering design study for a rocket-based 50 km injection system and associated sulfate aerosol production is warranted.

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

**Acknowledgments**

The CESM project is supported by the National Science Foundation and the Office of Science (BER) of the U.S. Department of Energy. I acknowledge the Global Model Simulation System Platform and High Performance Computing Public Service Platform of Jinan University.

This work is supported by the National Key R&D Plan of China (2024YFF0808501), National Natural Science Foundation of China (42121004) and the second Tibetan Plateau Scientific Expedition and Research Program (STEP, 2019QZKK0604).

**Data availability**

The model simulations can be downloaded via https://doi.org/10.17605/OSF.IO/5F6HU. AVHRR observations are publicly available at https://www.earthdata.nasa.gov/sensors/avhrr. The GloSSAC merged data can be downloaded via https://asdc.larc.nasa.gov/project/GloSSAC/GloSSAC_2.22.

**Author contribution**

Conceptualization: PY, YP

Methodology: PY, YP

Investigation: PY, YP, RWP, ST, OBT, RSG, KHR, MR, ER, JB

Visualization: YP, PY

Funding acquisition: PY

Project administration: PY

Supervision: PY

Writing – original draft: PY

Writing – review & editing: PY, OBT, KHR, RSG

**Competing interests:**

One author is a member of the editorial board of journal ACP.

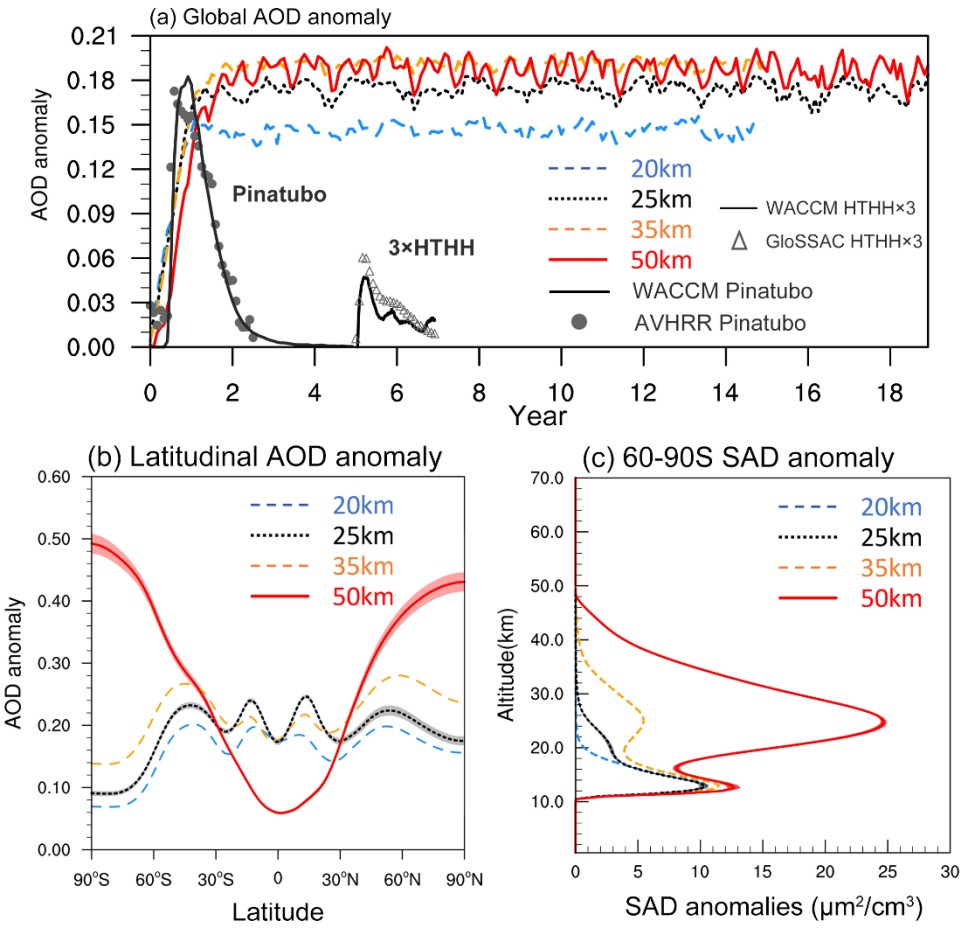

**Figure 1: Latitudinal and vertical distributions of aerosol in the 50-km injection scenario relative to the lower-altitude injection scenarios (20-km, 25-km, 35-km). (a) Simulated global and annual mean aerosol optical depth (AOD) anomalies from SAI scenarios with different injection altitudes. Simulated AOD anomalies following 1991 Pinatubo eruption and 2022 Hunga eruption (denoted by the solid black lines) are compared with the observations by AVHRR (denoted by the grey dots) (Zhao et al., 2013) and GloSSAC (denoted by the grey open triangles) (Kovilakam et al., 2020), respectively. For easier visualization, we multiply both of the observed and modeled AOD anomalies following HTHH eruption by a factor of three; (b) simulated latitudinal distribution of annual mean AOD anomaly from different SAI scenarios; (c) simulated vertical distributions of the Antarctic (60°S-90°S) aerosol surface area density anomaly from various injection scenarios averaged from September-October-November (SON). Shadings in panel b-c denote the simulated standard deviation of AOD or SAD in the 50 km and 25 km injection scenarios of the ensemble members. All scenarios inject SO₂ in two model grid points near (15°S,0°E) and (15°N,0°N) with a rate of 10 Tg per year (i.e. 5 Tg per year at each grid box).**

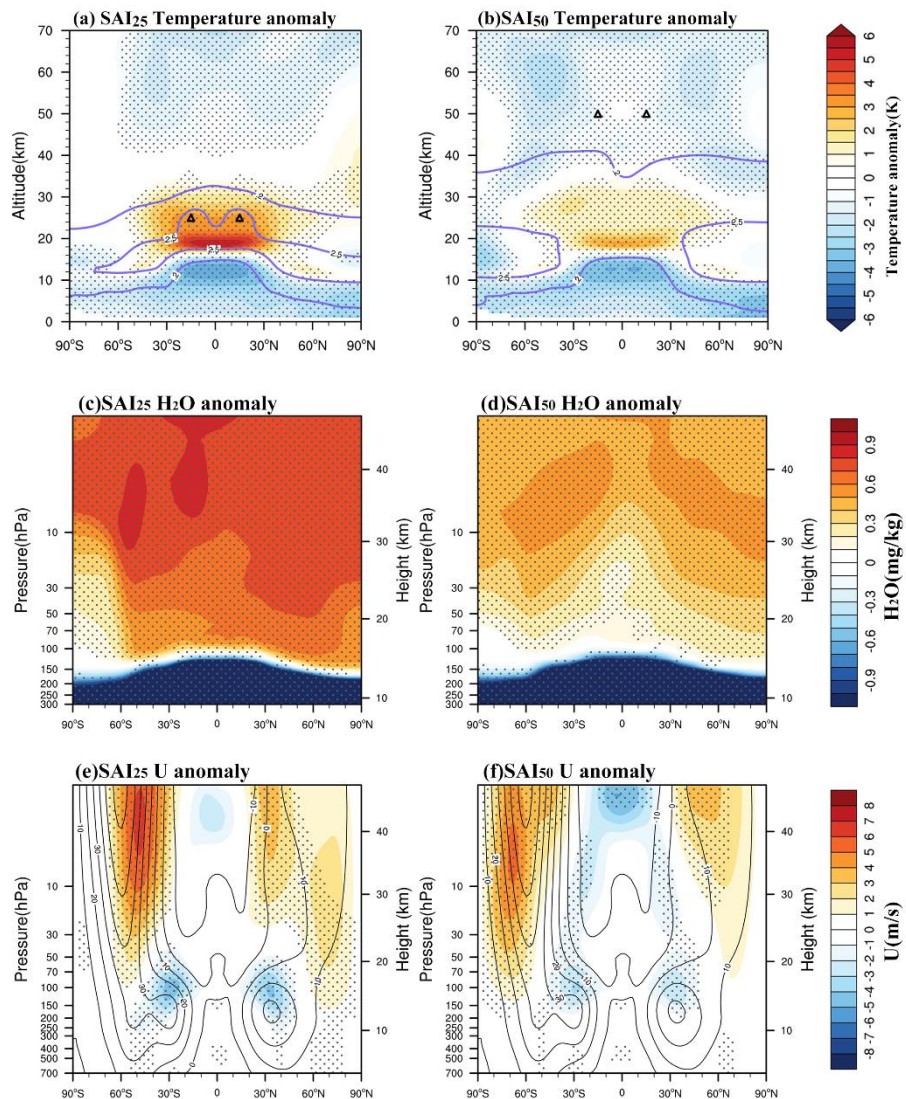

**Figure 2: Less stratospheric warming and more polar surface cooling in the 50-km injection scenario (SAI₅₀) relative to the 25-km injection scenario (SAI₂₅). (a) The vertical distribution of the zonal and annual mean temperature anomalies due to SAI₂₅. The simulated annual mean aerosol concentrations of 2.5 and 0.2 $\mu$g m⁻³ are denoted by the contour lines. The triangles denote the injection latitudes and altitudes; The stipple points denote the statistical significance at 95% level; (b) same as (a) but for SAI₅₀; (c-**

470 **d) same as (a-b) but for water vapor. (e-f) same as (a-b) but for zonal winds. The annual mean climatological zonal wind fields without SAI are shown in the contour lines.**

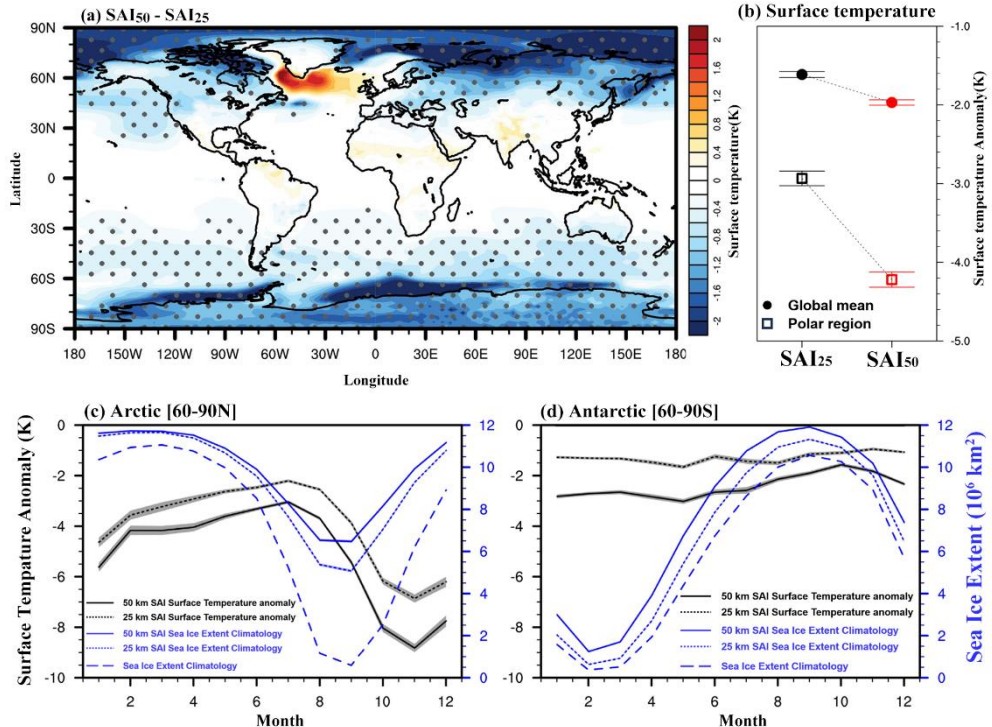

**Figure 3: Less stratospheric warming and more polar surface cooling in the 50-km injection scenario (SAI₅₀) relative to the 25-km injection scenario (SAI₂₅). (a) The difference in the simulated annual mean surface temperature response between SAI₅₀ and SAI₂₅. The stipple points denote the statistical significance at 95% level; (b) the simulated multi-annual and global mean temperature anomalies from SAI₂₅ and SAI₅₀ are denoted by the black and red filled dots, respectively. The temperature anomalies averaged in the polar regions are denoted in open squares. The error bars denote one standard deviation of the mean; (c) Simulated seasonal distributions of the surface temperature anomalies in Arctic (60-90°N) due to SAI₂₅ and SAI₅₀ are shown in the solid and dashed red lines, respectively (left Y-axis). The error bars denote simulated standard deviation of the mean from ensemble simulations. The simulated seasonal variance of sea ice extent in Arctic (60-90°N) from the control (i.e. no SAI), SAI₂₅, SAI₅₀ scenarios are denoted by the solid black, dashed blue and solid blue lines, respectively (right Y-axis); (d) same as (c) but for Antarctic (60-90°S).**

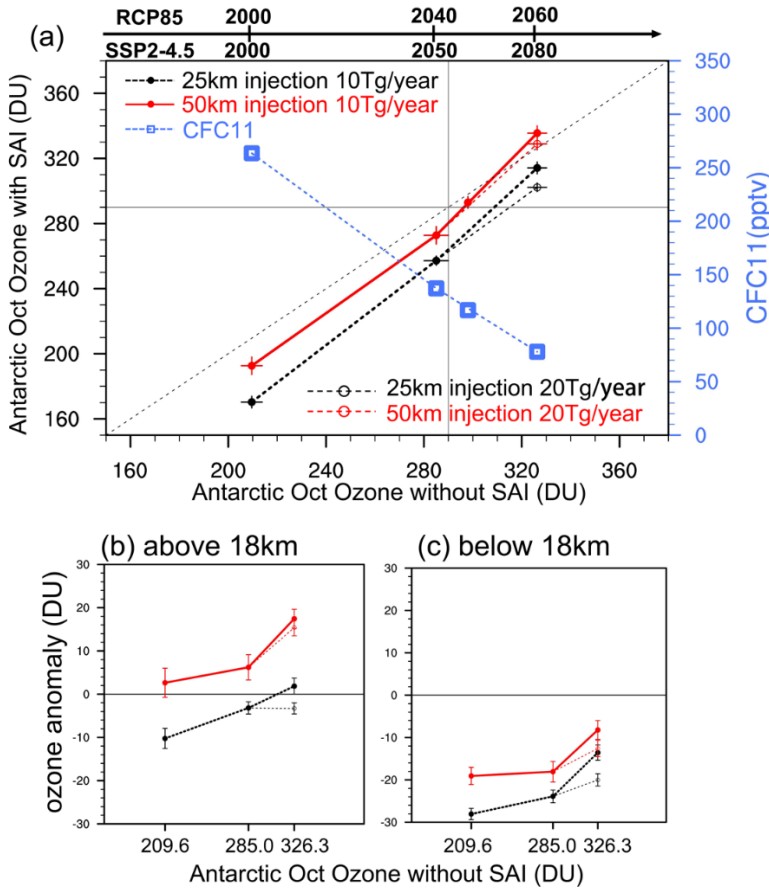

**Figure 4: Reduced Antarctic ozone depletion in the 50-km injection scenario (SAI₅₀) relative to the 25-km injection scenario (SAI₂₅). (a) Snapshot simulations of the October Antarctic (60°S-90°S) total column ozone (TCO) averaged over a period of 15 years with and without SAI when the global mean surface mixing ratios of CFC-11 are around 270, 135, 117 and 70 pptv, respectively. For each CFC-11 mixing ratio, a total amount of 10 Tg per year of SO₂ is injected into two model grid boxes at 15°S and 15°N, respectively. In addition, SAI with 20 Tg SO₂ injected per year is simulated when the CFC-11 mixing ratio is around 70 pptv. TCO with 25-km (50-km) SAI with 10 Tg per year injection is denoted by the black (red) filled circles. TCO with 20 Tg per year injection is denoted by the open circles instead. The error bars denote the standard deviation of the mean from the ensemble simulations with details in Materials and Methods. The global mean surface CFC-11 mixing ratios are denoted by the blue squares (right axis). The estimated timeline of TCO from the RCP8.5 (Tilmes et al., 2020) and SSP2-4.5 (Macmartin et al., 2022) scenarios without SAI are denoted in the upper part of the panel. The black dashed line denotes equal TCO between simulations with and without SAI. The TCO value in 1980 (i.e. ~290 DU) is denoted by the thin black line; (b) simulated October Antarctic column ozone anomalies above 18 km with variant mixing ratios of CFC-11. The simulated TCO anomalies with SAI injections (10 Tg per year) at 25-km and 50-km are denoted by the black and red filled circles, respectively. TCO anomalies with the injection rate of 20 Tg per year are denoted by the open circles. The error bars denote the simulated 1σ uncertainty from the ensemble simulations; (c) same as (b) but for the column ozone anomalies from the ground to 18 km.**

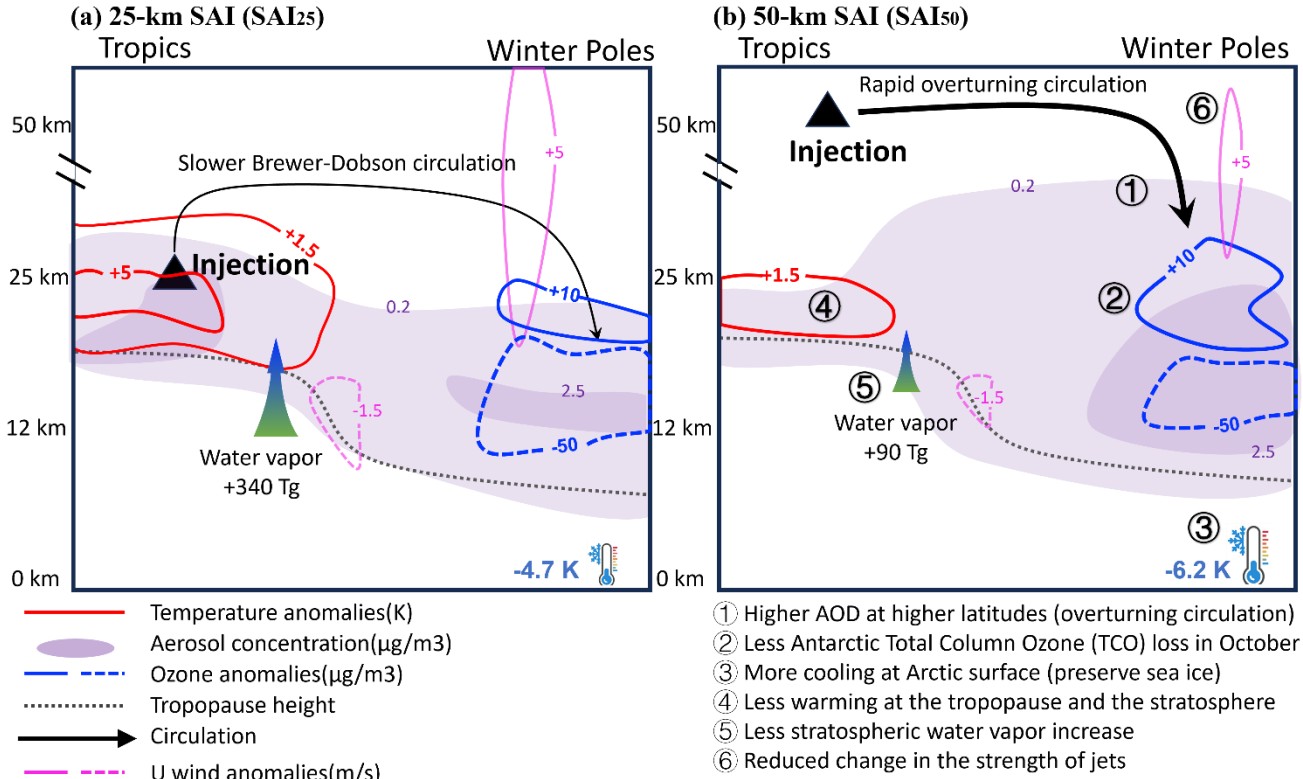

**(a) 25-km SAI (SAI25)**

Tropics — Winter Poles

50 km
Slower Brewer-Dobson circulation
+5
+1.5
25 km  +5  ▲ Injection
0.2  +10
2.5
12 km  -1.5  -50
Water vapor +340 Tg
0 km  **-4.7 K**

**(b) 50-km SAI (SAI50)**

Tropics — Winter Poles

Rapid overturning circulation
50 km  ▲ Injection  ⑥  +5
0.2  ①
+10
25 km  +1.5  ②
④
5  -1.5
Water vapor +90 Tg
12 km  -50  2.5
③  **-6.2 K**

Legend:
- Temperature anomalies(K) — red solid line
- Aerosol concentration(µg/m3) — purple shaded
- Ozone anomalies(µg/m3) — blue dashed line
- Tropopause height — dotted line
- Circulation — black arrow
- U wind anomalies(m/s) — pink dashed line

① Higher AOD at higher latitudes (overturning circulation)
② Less Antarctic Total Column Ozone (TCO) loss in October
③ More cooling at Arctic surface (preserve sea ice)
④ Less warming at the tropopause and the stratosphere
⑤ Less stratospheric water vapor increase
⑥ Reduced change in the strength of jets

**Figure 5: A schematic depiction comparing the aerosol distribution, polar ozone depletion and other climate impacts between the 25-km injection (SAI25, panel a) and the 50-km injection (SAI50, panel b). The climate impacts include tropopause warming, surface cooling, change of the strength of the subtropical and polar jets and water vapor transport into the stratosphere. The anomalies of the annual mean aerosol concentration, tropopause temperature, water vapor, zonal wind, and the October Antarctic ozone concentrations labeled in the plot are derived from the SAI simulations with injection rate of 10 Tg $SO_2$ per year under the**

**ODS conditions of year 2040. The climate advantages of SAI50 relative to SAI25 are numbered and listed below the right panel.**
