# Peer review of "Injection Near the Stratopause Mitigates the Stratospheric Side Effects of Sulfur-Based Climate Intervention"

_EGUsphere, 2025_

## Author Comment (AC3)

**Response to Reviewers' Comments**

Injection Near the Stratopause Minimizes the Stratospheric Side Effects of Sulfur-Based Climate Intervention

Pengfei Yu[1]*, Yifeng Peng[2], Karen H. Rosenlof[3], Ru-Shan Gao[3,4], Robert W. Portmann[3], Martin Ross[5], Eric Ray[3,4], Jianchun Bian[6], Simone Tilmes[7] and Owen B. Toon[8]

We thank very much the reviewers for their helpful comments. The response to each reviewer's comment is marked in blue.

**RC1**

The authors present an analysis of the effects of stratospheric aerosol injections in the upper stratosphere, in comparison to more typical lower stratospheric injections, as simulated in the WACCM-MAM3 climate-chemistry-aerosol model. While the results are generally presented in a clear way and could be interesting I think that the analysis lacks depth in several respects. My major concerns are listed below.

The comparison of emissions at 50 km is only done with respect to equatorial emissions at lower altitudes. However, there are earlier studies comparing different lower tropospheric emission strategies and also reporting some benefits in comparison to equatorial emissions. I think the authors need to put the potential benefits of their strategy into the perspective of these other emission strategies.

We assume the reviewer meant to ask the difference among various lower stratospheric (not "tropospheric") injection strategies.

We added discussions about high-latitude injection strategy in Lines 56-62:

"Tropical injection leverages the ascending branch of the BDC to efficiently transport aerosols into the global stratosphere, producing cooling across hemispheres and more effective surface cooling, while equatorial injection leads to substantial overcooling in the tropics and residual surface warming in the high latitudes (Kravitz et al., 2019; Tilmes et al., 2018b). High-latitude injections reduce the stratospheric warming, enhance polar cooling and sea ice preservation compared with tropical injection strategies (Lee et al., 2021; 2023b). However, it requires larger

injection amounts to achieve the same global cooling as tropical injections due to the shorter aerosol lifetime (Henry et al., 2024; Zhang et al., 2024; Duffey et al., 2025)."

I think that the model validation in 3.1 is too superficial concerning the upper mesosphere. I agree that it is useful to show an AOD comparison for the Hunga Tonga eruption where the sulfate also reached the upper troposphere. However, a distinct difference between emissions at 50 km and lower altitude emissions is the reported meridional distribution. This will depend on the representation of the overturning circulation in the upper stratosphere which for which an important feature is the semiannual oscillation near the stratopause. How well is that represented in the model. I don't think it is sufficient to say "temperature at 100 hPa, QBO strength, and polar vortex strength" show reasonable agreement with reanalysis data, because these are all features evaluated in the lower to middle stratosphere. Of course, evaluations of near-stratopause circulation are more difficult due to the lack of observations, but I think this needs to be discussed.

We agree that model evaluations near the stratopause are not easy due to a lack of observations. We compare the simulated Semiannual Oscillation (SAO) with reanalysis data in Figure S1c to evaluate model performance near the stratopause. The model can reproduce the 1 mb winds from MERRA2. We discuss SAO validation in Lines 124:

"Comparison with MERRA2 reanalysis data (2000-2020) shows reasonable agreement in key stratospheric metrics including temperature at 100 hPa, Quasi-biennial Oscillation (QBO) strength, Semiannual Oscillation (SAO) strength (1 mb tropical zonal winds) and polar vortex strength (Fig. S1), ...."

[Figure]

Figure S1: Box plots comparing stratospheric variability between model simulations and MERRA2 reanalysis (2000-2020). (a) near-global (60°S-60°N) temperatures at 100 mb; (b) Quasi-biennial Oscillation (QBO) strength (tropical zonal mean zonal wind at 50 mb) with east phase (dashed) and

west phase (solid); (c) Semiannual Oscillation (SAO) strength (tropical zonal mean zonal wind at 1 mb) with east phase (dashed) and west phase (solid); (d) Antarctic (solid) and Arctic (dashed) polar vortex strength (denoted by the geopotential height averaged over 65°S-90°S at 50 mb) in the model and MERRA2 reanalysis dataset between year 2000 and 2020. Box plots show the median (horizontal line), 25th-75th percentiles (box), and 5th-95th percentiles (whiskers).

I disagree with the title. The study doesn't provide evidence that SAI emissions near the stratopause "minimize" side effects. There could be other strategies reducing the side effects further that haven't been tested. Who knows what effects would be caused by an emission in the upper mesosphere?

Revised to be: "Injection Near the Stratopause Mitigates the Stratospheric Side Effects of Sulfur-Based Climate Intervention"

At the end of Section 3.2 the authors write "The distinct latitudinal and vertical distributions of aerosols in SAI50 enhance climate cooling benefits while minimizing negative impacts of climate intervention." Besides the issue with the word "minimizing" mentioned in connection with the title, I also think this statement is not sufficiently backed up, at least not at this point of the manuscript. Possibly the sentence is meant as an announcement for the following two subsections, but it sounds like a summary.

We revised the sentence in Line 156: "The distinct latitudinal and vertical distributions of aerosols in $SAI_{50}$ are expected to influence the climate cooling benefits and mitigate the associated stratospheric impacts, as detailed in the following subsections."

More in general the manuscript suffers from the lack of the definition of a goal for the SAI. Without such a goal, without defining a metric it is impossible to compare which strategy performs best. The goal could (but doesn't have to) be to produce a climate as similar as possible to an unengineered climate of the same global temperature at lower greenhouse gas levels. In this sense, it is not clear if the stronger Arctic amplification simulated for $SAI_{50}$ than for $SAI_{25}$ is actually a desired effect. How strong is the Arctic amplification in greenhouse gas caused warming in WACCM? Which injection strategy is counteracting the amplification more exactly?

Thanks for the comment.

We acknowledge the importance of defining clear metrics for evaluating SAI strategies. Our study uses a fixed injection rate of 10 Tg per year as an idealized experimental design to compare the fundamental differences in climate response between $SAI_{25}$ and $SAI_{50}$. This simplified approach

allows us to isolate and understand the physical mechanisms controlling aerosol transport and distribution at different injection heights.

In more practical scenarios (e.g. GLENS, ARISE, GeoMIP), injection amounts would need to vary over time to counteract increasing greenhouse gas forcing (Henry et al., 2024; Macmartin et al., 2022; Tilmes et al., 2018b). Regarding Arctic amplification, previous studies have shown that it cannot be fully offset under tropical SAI scenarios at ~25 km (Figure R1, Henry et al., 2024). While SAI$_{25}$ effectively offsets greenhouse gas–induced warming in the tropics and mid-latitudes, the Arctic still experiences significant residual warming. The enhanced high-latitude cooling in SAI$_{50}$ could potentially provide better compensation for Arctic amplification.

[Figure]

Figure R1. The ensemble-mean temperature change in 2050-2069 relative to the target period (2014–2033) in SSP2-4.5. (Figure A6 in Henry et al., 2024). (a) The surface temperature anomalies for SSP2-4.5; (b) The surface temperature anomalies for tropical SAI strategy (15S/15N).

We add the discussions in the manuscript in Line230-235:
"Additionally, while this study uses idealized fixed-rate injections to compare fundamental differences between injection heights, more practical implementation would require varying injection rates to meet specific climate objectives (Henry et al., 2024; Macmartin et al., 2022; Tilmes et al., 2018b). The enhanced high-latitude cooling observed in SAI$_{50}$ suggests potential advantages for offsetting Arctic amplification, though determining optimal injection strategies would depend on defined climate goals and metrics."

Concerning Arctic amplification, the authors write: "In SAI50, the simulated 22% greater global mean surface cooling compared to the 10% increase in global mean AOD (Fig. 1a), is primarily attributed to Arctic amplification effects (Barnes and Polvani, 2015), with a minor contribution from the reduced stratospheric water vapor enrichment (Fig. 2c-d)." I don't understand this statement. Arctic amplification, depending on the mechanism which causes it in WACCM, should be part of the temperature response in both strategies. Shouldn't part of the difference between SAI50 and SAI25 be due to the different aerosol distributions. Is the idea that polar aerosols create a larger forcing than low-latitude aerosols? Would that be related to the surface-temperature

dependence of stratospheric aerosol forcing as discussed by Hegde et al. (2025). Or to aerosol forcing being more efficient at high than low latitudes? Anyhow, I think it is necessary to physically explain the relatively strong additional global cooling for a relatively weak AOD increase.

Sorry for the confusion. The reviewer is correct that Arctic amplification occurs in both $SAI_{50}$ and $SAI_{25}$ scenarios. The stronger cooling response in $SAI_{50}$ (22% greater global mean surface cooling) relative to its AOD increase (10%) can be attributed to two factors:

- The spatial distribution of aerosols: $SAI_{50}$ has a higher proportion of aerosols at high latitudes compared to $SAI_{25}$.

- Enhanced efficiency of aerosol forcing at high latitudes: Due to Arctic amplification mechanisms (such as ice-albedo feedback and stable atmospheric conditions), aerosol forcing in the Arctic region produces a stronger cooling effect per unit AOD than at lower latitudes.

This explains why a relatively modest 10% increase in global mean AOD results in a disproportionate 22% enhancement in global mean surface cooling in $SAI_{50}$.

We clarified in the manuscript in Lines 193-197:

"In $SAI_{50}$, the simulated 22% greater global mean surface cooling compared to the 10% increase in global mean AOD (Fig. 1a) primarily reflects the higher proportion of aerosols distributed at high latitudes, where Arctic amplification mechanisms enhance the cooling efficiency of aerosol forcing. Arctic amplification processes, including ice-albedo feedback and stable atmospheric conditions (Barnes and Polvani, 2015), contribute to this enhanced regional cooling response. A minor contribution also comes from the reduced stratospheric water vapor enrichment (Fig. 2c-d)."

Figures 3c and 3d show simulated annual cycles of the high-latitude cooling signals. In the Arctic there is a pronounced seasonal cycle, while it is negligible in the Antarctic. This behaviour is just stated but not explained. To develop trust in such signals it is important to explain the physical mechanism causing this difference. Moreover, with respect of the "cooling benefits" discussion it would be important to discuss if these different annual cycles just offset different annual cycles of high-latitude greenhouse gas warming or if seasonal cycles are strongly modified.

The pronounced seasonal cycle of Arctic cooling compared to Antarctic cooling under SAI scenarios reflects fundamental differences in surface characteristics between these regions. The Arctic Ocean's seasonal sea ice plays a crucial role, while Antarctica's permanent ice sheet leads to more stable conditions year-round.

Arctic amplification - the enhanced temperature response in the Arctic region - occurs primarily from October to April and is strongly tied to sea ice loss (Dai et al., 2019). During summer, incoming solar energy is consumed by sea ice melt. In fall-winter, areas where sea ice has

retreated expose open water, leading to increased outgoing longwave radiation and heat fluxes that drive stronger temperature changes. This mechanism explains why both Arctic warming under greenhouse gas forcing and Arctic cooling under SAI scenarios show maximum intensity during October-April.

We have explained this mechanism in the manuscript (Lines 206-209):

"The Arctic cooling exhibits pronounced seasonality, with maximum effects during fall-winter seasons (Fig. 3c). This seasonal pattern aligns with the mechanism of Arctic amplification, which is driven by increased outgoing longwave radiation and heat fluxes from areas of seasonal sea ice loss during October-April (Dai et al., 2019). In contrast, Antarctica's year-round ice cover results in more uniform cooling throughout the year (Fig. 3d)."

SAI effectively offsets the seasonal peak of polar warming, while the simulated seasonal cycles of surface temperature over both poles remain largely unchanged, as shown in Fig. R2.

[Figure]

Figure R2: (a) The averaged surface temperature over Arctic (60N-90N). The black line denotes the global warming scenario in 2040. The blue and red lines denote the SAI25 and SAI50 scenarios, respectively; (b) Same as (a) but for Antarctic over 60S-90S.

Finally I see a major issue with the lack of discussion of the additional costs of emitting near the stratopause compared to the lower troposphere. The authors are briefly mentioning the option of using rockets and conclude that the "results clearly indicate that a detailed engineering design study […] is warranted." I think at least a brief estimation of costs based on existing rockets would be necessary. One could argue that scientifically it is interesting to see the dependence of SAI effects on the injection height. But if feasibility plays no role, why not emit at 70 or 100 km? As the authors claim to "propose a novel SAI approach" I think a minimum effort on estimating feasibility is necessary.

We appreciate the reviewer's comment regarding implementation costs. While cost considerations are indeed important for real-world deployment of any SAI approach, our study represents a theoretical exploration using idealized numerical experiments and focuses specifically on

understanding the physical mechanisms and climate response to stratospheric aerosol injection at different altitudes. This theoretical exploration is essential for advancing our fundamental understanding of stratospheric dynamics and aerosol-climate interactions.

While we acknowledge that implementation feasibility is an important consideration for any proposed climate intervention strategy, a detailed engineering and economic analysis would require expertise beyond atmospheric sciences and would constitute a separate study entirely. Our results provide the physical basis necessary for future interdisciplinary assessments that could then evaluate technical feasibility and economic viability in detail.

---

## Author Comment (AC4)

**Response to Reviewers' Comments**

Injection Near the Stratopause Minimizes the Stratospheric Side Effects of Sulfur-Based Climate Intervention

Pengfei Yu[1]*, Yifeng Peng[2], Karen H. Rosenlof[3], Ru-Shan Gao[3,4], Robert W. Portmann[3], Martin Ross[5], Eric Ray[3,4], Jianchun Bian[6], Simone Tilmes[7] and Owen B. Toon[8]

We thank very much the reviewers for their helpful comments. The response to each reviewer's comment is marked in blue.

**RC2**

Review by Thomas Peter

General comments

This is an interesting manuscript on a new idea for how climate intervention through stratospheric aerosol injection (SAI) could be implemented by injecting at much higher altitudes (~50 km) than previously proposed. This could lead to significantly less harmful side effects than previous proposals to inject SO2 into the lower stratosphere. This could reduce both the strong warming of the lower stratosphere due to IR absorption by H2SO4-H2O aerosol, which affects climate and precipitation zones in the troposphere, and the depletion of stratospheric ozone. As the authors describe in their manuscript, this new idea could even boost the efficiency of surface cooling.

To my knowledge, this idea is original, and the proposed method could potentially be very important and would fit well into ACP. The authors are to be commended for developing this concept. Unfortunately, however, I do not believe that the technical details necessary to justify the feasibility of this novel method are sufficiently developed for publication in ACP.

Thanks Prof. Thomas Peter for the constructive suggestions, which help to improve the manuscript. Point-to-point responses are marked in blue.

The differences in aerosol formation at an altitude of 50 km compared to 25 km are considerable and would need to be discussed:

(1) The temperatures are so high that the formation of H2SO4-H2O droplets close to injection altitude seems unlikely.

(2) The air density is so low that the fall velocity of particles, if they form, will be very high, which is a major limitation.

(3) The H2SO4 photolysis, which is neglected in this modeling, could massively alter the model results.

These three aspects are not addressed in the submitted manuscript (with the exception of a reference to the fact that H2SO4 photolysis is neglected).    However, the issues associated with these aspects are central to such a new proposal and must not be ignored.    Therefore, I do not think that the manuscript can be accepted for publication in its present form.

Sorry for the confusion. We think all three aspects stem from one potential misunderstanding that needs clarification:

We inject $SO_2$ at 50 km, but our simulations show that sulfate aerosol formation occurs at much lower altitudes. Figure R3 shows the simulated zonal and vertical distributions of $SO_2$, sulfate aerosol, and $H_2SO_4$ anomalies in both $SAI_{25}$ and $SAI_{50}$ scenarios. In $SAI_{25}$, the simulated aerosol peak is located around 25 km in the tropics (the injection region). However, in $SAI_{50}$, the sulfate aerosol forms at lower altitudes, with peak concentrations below 20 km at high latitudes. The simulated $H_2SO_4$ anomalies are several orders of magnitude smaller than $SO_2$ at 50 km. We chose to inject $SO_2$ at 50 km because the overturning circulation rapidly transports these precursor gases poleward, leading to a more uniform global distribution. In the revised manuscript, we add Figure R3 into the supplement.

[Figure]

Figure R3: (a) The vertical distribution of the zonal and annual mean $SO_2$ anomalies in $SAI_{25}$. (b) same as (a) but for $SAI_{50}$; (c-d) same as (a-b) but for sulfate aerosol. (e-f) same as (a-b) but for $H_2SO_4$. Note that the contour range in panel (e-f) $H_2SO_4$ is 3 magnitudes smaller than sulfate and $SO_2$.

Detailed responses to each specific aspect are shown below in the Specific Comments section.

Specific comments

In the following, I will first explain my concerns regarding these three points in more detail and then provide a list of minor comments and suggestions for improvement of the manuscript.

Regarding (1):   At the stratopause, temperatures range between 260 and 270 K.   At such high temperatures, the normal Junge aerosol can no longer exist.   The H2SO4 vapor pressure of aqueous sulfuric acid is approximately 10-8 hPa (see Fig. 4 of Carslaw et al., Revs. Geophys., 35, 2, p. 125,1997).   At the stratopause this would correspond to 10 ppbv H2SO4 in the gas phase, i.e. about 100 times the total mixing ratio normally present in the lower stratosphere during volcanically quiescent periods.   I think the mixing ratio reached by SAI50 would remain lower.   Therefore, the nucleation of sulfuric acid particles probably only begins well below the injection height.   Figure 1C seems to confirm that.   How does the model treat this nucleation?   No information is provided about the microphysical code, which for this proposal should be of major concern.   I suppose the microphysics is treated by a modal approach, and it would be important to see size distributions at various altitudes.

The reviewer is correct that the formation of sulfate aerosol is difficult at stratopause near 50 km. As shown in Fig.R2, the simulated sulfur-species at 50 km in $SAI_{50}$ scenario is dominated by $SO_2$, rather than $H_2SO_4$ gas nor sulfate aerosols. The simulated $H_2SO_4$ gases is 3 orders of magnitude lower than $SO_2$ at 50 km in $SAI_{50}$. Aerosols form when $SO_2$ gases are transported to higher latitudes and lower altitudes, with peak aerosol concentrations simulated in lower-middle stratosphere (Fig.R3). We added Figure R3 to supplement file (Figure S3), and explained this point in Line 151:

"Note that the sulfate aerosol evaporates into sulfuric acid gas above 35-40 km but reforms when the gas is transported to lower altitudes (10-30 km) via large-scale circulation."

In Line 138-141 :

"…It's important to note that while $SO_2$ is injected at 50 km, the actual sulfate aerosol formation occurs at much lower altitudes (primarily between 10-30 km) due to the rapid transport of precursor gases and more favorable conditions for aerosol formation at lower altitudes. Above 40 km, the simulated stratospheric sulfur species primarily exist in the form of SO2, with ~3 orders of

magnitudes higher than H2SO4 (Fig. S3). Above 40 km, the simulated stratospheric sulfur species primarily exist in the form of $SO_2$, with ~3 orders of magnitudes higher than $H_2SO_4$ (Fig. S3)."

We also describe aerosol scheme MAM3 in the method section in Line78-81:

"MAM3 provides a physically-based treatment of aerosol size, mixing, and key microphysical processes, including nucleation, growth, deposition, and interactions with clouds and precipitation (Liu et al., 2012). The nucleation of sulfate aerosol is produced from aqueous-phase $SO_2$ oxidation and to a lesser extent from $H_2SO_4$ condensation on pre-existing aerosol (Liu et al., 2012)."

Regarding (2): The air is so thin at the injection height that particles with a radius of 100 nm sediment by about 10 km within a month (eyeballed from Fig. 2 of Müller & Peter, Ber. Bunsenges. Phys. Chem. 96, p. 353, 1992). Since this is a fundamental aspect of the proposed injection scheme, it would be necessary to check the model's implementation of this process carefully and to provide arguments, why this fast sedimentation does not invalidate the whole procedure.

We agree that aerosol sedimentation is faster in thinner air at higher altitudes. However, in our $SAI_{50}$ simulations, the injected aerosols do not remain near 50 km but accumulate around 25–30 km altitude (see Fig. 1c).

We also confirm that CESM-WACCM can represent gravitational settling, and this process has been validated against observational estimates of aerosol lifetimes from major volcanic eruptions, such as Pinatubo and Hunga Tonga (Figure 1a).

Regarding (3): The photolysis of H2SO4 molecules is a central process in this scheme and cannot be ignored without good reason. A quick test with our own chemistry-climate model shows that the amount of condensed H2SO4 in the Junge aerosol layer is 2-4 times higher without photolysis than with photolysis. I would estimate that the reduction of aerosol mass in the upper stratosphere could more than a factor of 10. I must therefore assume that in the modeling work shown, a significant portion of the aerosol is formed solely due to the lack of photolysis in the model.

Following the reviewer's suggestion, we compared the vertical distributions of sulfur-containing species in our SAI simulations. As shown in Figure R3-4, the simulated concentrations of $H_2SO_4$ are 1-3 orders of magnitude lower than those of sulfate aerosols.

We add $H_2SO_4$ photolysis into the model and compare the vertical distributions of the sulfur species with and without photolysis. Shown in Fig. R3b, the simulated vertical distributions of sulfur-containing gases and aerosols remain largely unchanged between simulations with and without photolysis. These results indicate that $H_2SO_4$ photochemistry has a limited impact on the overall sulfur distribution in $SAI_{50}$ mainly because sulfate aerosols form in much lower altitude

instead of stratopause, where the precursors SO$_2$ are transported rapidly by the overturning circulation.

[Figure]

Figure R4. (a) Simulated vertical distributions of the Antarctic SAD with (black) and without (red) H$_2$SO$_4$ photolysis (same as Fig 1c red line) averaged from September-October-November (SON). (b) Simulated annual global mean vertical distribution of the sulfur species with (black) and without (red) H$_2$SO$_4$ photolysis. The simulated sulfate, H$_2$SO$_4$, and SO$_2$ are denoted by solid, dashed, and dotted line, respectively.

While points (1) and (2) can probably be positively resolved with the existing model runs (showing that the model correctly calculates the partial and vapor pressures of the aqueous sulfuric acid under upper stratospheric conditions and the sedimentation velocity of the aerosol particles formed), point (3) is likely to pose a bigger problem. If photolysis cannot be incorporated into the existing model, at least a very clear warning should be included for the reader that this omission may significantly influence the result and diminish the efficiency of the proposed method.

Please see the point-to-point response above.

Technical comments listed by line number

1.  L. 19: "SAI using sulfur cools the planet" à "SAI using sulfur has been proposed to cool the planet"

    Done

2. L, 20: Better don't talk about "traditional" SAI, in particular not in the first sentence of the abstract. All SAI is still in the proposal phase and largely unsubstantiated ideas, nothing traditional.

Revised to be "A commonly proposed SAI, with sulfur dioxide injection rate of 10 Tg/year at 25 km …."

3. L. 38: In addition to Ferraro et al. (2015) and Visioni et al. (2021), another excellent example making this point is Wunderlin et al., "Side effects of sulfur-based geoengineering due to absorptivity of sulfate aerosols", GRL, 2024.

Done

4. L. 66: I do not understand the "positive ozone chemical tendency".

We rewrite the sentence in Line 68: "By doing so, high-altitude sulfate aerosols reduce NOx levels, slowing NOx-driven ozone loss and allowing ozone to accumulate in the middle stratosphere, which can offset the ozone loss caused by reactive halogen species in the lower stratosphere."

5. L. 73: Here I expected more information on the type and characteristics of the microphysical module used in this modelling work

We also describe aerosol scheme MAM3, in Lines 78-81:

"MAM3 provides a physically-based treatment of aerosol size, mixing, and key microphysical processes, including nucleation, growth, deposition, and interactions with clouds and precipitation (Liu et al., 2012). The nucleation of sulfate aerosol is produced from aqueous-phase $SO_2$ oxidation and to a lesser extent from $H_2SO_4$ condensation on pre-existing aerosol (Liu et al., 2012)."

6. L. 79: I do not understand and do not accept that the fact that the column-integrated stratospheric burden of H2SO4 is much smaller that the burden of sulfate aerosols could be used for not having to take the photochemistry of H2SO4 into account. In the warm upper stratosphere, all H2SO4 is gaseous and exposed to H2SO4 + hv.

As discussed in previous response, the simulated aerosol peaks in 20-25 km in altitude instead of 50 km. $SO_2$ is released at 50 km but transported polewards and downwards until sulfate aerosols are formed.

We delete the sentence in the method section and address the $H_2SO_4$ photochemistry in the result section in Line 141-142:

"…, and the simulated stratospheric sulfur species primarily exist in the form of sulfate and $SO_2$, with ~3 orders of magnitudes higher than $H_2SO_4$ (Fig. S3)

7. L. 84: "SO2 was continuously injected … with a total rate of 10 Tg per year". It might be more meaningful to say "with a total rate of 27 Mg per year" to stress the continuous character, or even "with a total rate of 27 tons per year".

In the SAI experiments, SO₂ was continuously and uniformly injected throughout the year, with a total annual amount of 10 Tg injected per year.

8. L. 87: 5 years of model spin-up plus 15 years of actual model run. This is okay.    But then, does Figure 1 show the 5 years of spin-up plus 10 years of model run?

In the revised manuscript, Figure 1a is extended to 20 years, consistent with 15 years model run with 5 years spin-up.

9. L. 90: I do not understand these scaling factors. Where are they from?

We rewrite in Line 99-101: "For simulations of year 2000, model is initialized with atmospheric ODS and Greenhouse Gases (GHGs) conditions of year 2000. For simulations of year 2040 (2065), the ODS and GHGs are fixed in the year of 2040 (2065)."

10. L. 103: The "2022 Hunga volcanic eruption". Okay, everybody knows which volcano this is, yet it is a pretty crude abbreviate

Revised to be "Hunga Tonga-Hunga Ha'apai"

11. L. 110: The "spread is designed to capture" sounds weird. Are you designing a spread or is the spread the result of your simulations?

Thanks. Revised to be: "The spread across our simulations of 45 ensemble members represents the natural variations in stratospheric circulation."

12. L. 127: Sentence confusing. For clarity, I would rewrite "… are similar for all lower altitude injections (at 20 km, 25 km and 35 km),...".

Done

13. L. 129: The word anomaly is used abundantly, also when it is just a "change" or even a total number. For instance, in Figure 1a is it really AOD anomaly or just AOD.    And why is it in Figure 1C simply "SAD", and not "SAD anomaly"?

Thanks for the comment. Figure 1a, b, c are all AOD or SAD anomalies. We modified Fig.1c label to be "SAD anomaly". We checked throughout the manuscript.

14. L. 260: "would be" instead "is".

Corrected

15. L. 295: Where is the dip in SAD (Fig. 1c) at 18 km come from?

We show the monthly vertical distribution of aerosols and found that the dip in SAD at ~18 km results from the interaction between transport and microphysical processes. Analysis of monthly vertical distributions reveals two distinct aerosol peaks:

- An upper peak (25-30 km) formed by continuous poleward transport of newly formed sulfate aerosols via the upper branch of the Brewer-Dobson Circulation (BDC)=

- A lower peak (~15 km) representing older aerosols that have descended through gravitational settling

During Antarctic spring (shown in Fig.1c), these two separate peaks create the observed dip at ~18 km. This pattern emerges as the BDC continuously brings new aerosols to high altitudes while previously transported aerosols settle to lower levels

[Figure]

Figure R5. Simulated monthly vertical distribution of SAD anomalies in $SAI_{50}$ from January (a) to December (l) after injection.

16. L. 295: The red curve in Fig. 1c would probably look quite different if H2SO4 + hv was taken into account.

We have conducted additional simulations that include H2SO4 photolysis (H2SO4+hv). The simulated Surface Area Density (SAD) vertical distributions are shown in Figure R3. While H2SO4 photolysis does affect the sulfate aerosol lifecycle, the overall vertical distribution pattern remains similar to our original simulations without photochemistry -

the simulated SAD still peaks between 20-30 km. This is because the primary formation of sulfate aerosol occurs at lower altitudes, and photolysis becomes more significant only at higher altitudes where the aerosol concentration is already low.

17. L. 296: Why distinct?

Deleted

18. L. 301: "Multiple" is not a verb.

Corrected

19. L. 305: "from ensembles" is slang. "of the ensemble members" would be appropriate.

Corrected

---

## Referee Report (RR1)

**Injection Near the Stratopause Minimizes the Stratospheric Side Effects of Sulfur-Based Climate Intervention**

by Pengfei Yu, Yifeng Peng, Karen H. Rosenlof, Ru-Shan Gao, Robert W. Portmann, Martin Ross, Eric Ray, Jianchun Bian, Simone Tilmes and Owen B. Toon

Revised manuscript October 2025

Review by Thomas Peter

The authors have largely addressed my concerns, but some points need further clarification.

In my first review, I had the following three main concerns:

(1) From the original manuscript, I got the impression that the $H_2SO_4$-$H_2O$ droplets proposed for SAI formed at the stratopause, even though temperatures at these altitudes are so high that the formation of $H_2SO_4$-$H_2O$ droplets is unlikely.
(2) Based on this impression, I criticized that the initial fall velocity of the particles would be very high, which poses a major complication for the model that would need to be addressed.
(3) Finally, I believed that the photolysis of $H_2SO_4$, which was neglected in this modeling, could massively alter the model results.

In the revised version, the authors address these three concerns as well as all individual comments.

Concerns (1) and (2):

The authors clarify that my impression that the $H_2SO_4$-$H_2O$ droplets in their model form at the stratopause was a misunderstanding. The "Results" section now states: "It's important to note that while $SO_2$ is injected at 50 km, the actual sulfate aerosol formation occurs at much lower altitudes (primarily between 10-30 km) due to the rapid transport of precursor gases and more favorable conditions for aerosol formation at lower altitudes." This is also clearly illustrated by the new Figure S3 in the supplementary information. Thank you. This is indeed in line with my expectations and does not change the potential value of the proposed new scheme. It also essentially addresses my concerns regarding points (1) and (2).

However, the false impression has not been completely dispelled. The fifth sentence of the abstract states: "In SAI50, the mean meridional overturning circulation near the stratopause rapidly transports aerosols to mid-high latitudes…". This still sounds as if the circulation near the stratopause transports aerosol particles to higher latitudes, which is not the case. Rather, the circulation near the stratopause transports gaseous $SO_2$ to higher latitudes, from where the $SO_2$ (plus some already formed gaseous $H_2SO_4$) is transported further to lower altitudes, finally oxidized completely to $H_2SO_4$ and forming $H_2SO_4$-$H_2O$ droplets only below an altitude of 30–35 km (through bimolecular homogeneous nucleation or heterogeneous nucleation, e.g., on meteorite dust particles).

The confusion continues in the Introduction, with the following statements: "To minimize the Antarctic ozone loss, it is essential that some sulfate aerosols from the intervention remain at high altitudes in the polar stratosphere. By doing so, high-altitude sulfate aerosols reduce NOx levels… In addition, aerosols formed at higher altitudes are rapidly transported to the mid-high latitudes rather than accumulating in the tropical lower stratosphere." I cannot see that aerosol particles, which form at higher altitudes, are then "rapidly transported to mid-high latitudes". I think the new Fig. S3d suggests instead that the particles are already at high latitudes when they nucleate and do not need to be transported there. This description has the potential to mislead readers and should be improved before publication.

Concern (3):

In response to my concern that photolysis of $H_2SO_4$ must not be neglected as it could massively alter the results, the authors present simulations with and without $H_2SO_4$ photolysis. First, I am surprised that they can do this so easily, as I had assumed that the previous neglect was due to the model not containing $H_2SO_4$ photolysis. Since this is obviously not the case, I wonder why they did not show all results including $H_2SO_4$ photolysis right away. Second, I am even more surprised about their result, namely that $H_2SO_4$ photolysis is completely negligible. Because this contradicts my statement of a "massive" effect, I would have expected them to discuss possible reasons for this contradiction. As my statement referred to background conditions (non-SAI), we reran our CCM (SOCOL) with SAI and found that the large impact reduces to < 10 % in sulfate concentration in the center of the aerosol layer. This confirms the figure shown by the authors. Now, that they have demonstrated that $H_2SO_4$ photolysis indeed plays a negligible role under SAI conditions, I do not understand why the authors continue to write "Note that the photolysis of $H_2SO_4$ gas … is not included in the model." Unless readers refer to this review, they may ask themselves the same questions I did.

I agree with publication in ACP, provided that the misleading statements regarding the aerosol formation are corrected and the consequences of neglecting $H_2SO_4$ photolysis are mentioned.

---

## Author Response (AR2)

**Response to Reviewers' Comments**

Injection Near the Stratopause Minimizes the Stratospheric Side Effects of Sulfur-Based Climate Intervention

Pengfei Yu[1]*, Yifeng Peng[2], Karen H. Rosenlof[3], Ru-Shan Gao[3,4], Robert W. Portmann[3], Martin Ross[5], Eric Ray[3,4], Jianchun Bian[6], Simone Tilmes[7] and Owen B. Toon[8]

We thank very much the reviewers for their helpful suggestions. The response to each reviewer's comment is marked in blue.

**RC1**

I'd like to thank the authors for considering and responding my comments. In many ways I think the manuscript has improved. I'm still not fully convinced of some of the replies.

I had remarked that the comparison of emissions at 50 km is only done with respect to equatorial emissions at lower altitudes while other emission strategies for the lower stratosphere exist. In response the authors have added three sentences to the introduction in which they refer to high-latitude emission strategies, and list some deficits of equatorial emissions that could be alleviated, among them "stratospheric warming". Now the main conclusion of this new manuscript seems to be indicated by the title: "Injection Near the Stratopause Mitigates the Stratospheric Side Effects …". I think it is necessary to refer to alternative strategies not only in the introduction but also in the conclusions. How do the potential benefits of the stratopause emission strategies compare to other alternative strategies?

we added discussion in Line 275-278:

"Note that polar injection strategies also aim to mitigate tropical lower-stratospheric warming and preserve sea ice (Lee et al., 2021; Lee et al., 2023b), the SAI50 scenario requires less aerosol mass to achieve the same temperature target due to longer aerosol lifetime. In addition, the polar aerosol layer in SAI50 resides at higher altitudes than in polar injection scenario, which helps suppress NOx-catalyzed ozone loss and mitigate the severe ozone depletion caused by low-stratospheric aerosol accumulation."

I'm not convinced yet concerning the explanation for 22% greater cooling for 10% AOD increase in SAI50 compared lo lower stratospheric injection. The authors now write this "primarily reflects the higher proportion of aerosols distributed at high

latitudes" and a "minor contribution also comes from the reduced stratospheric water vapor enrichment". While I think this is possible, I don't understand how this was analyzed and why the forcing dependence on surface temperature can be excluded as a contributor.

we rewrote in Line 187-195:

"In SAI50, the simulated 22% greater global mean surface cooling compared to the 10% increase in global mean AOD (Fig. 1a) reflects a combination of factors, including a higher proportion of aerosols distributed at high latitudes that enhances the efficiency of aerosol forcing through Arctic amplification processes, such as ice-albedo feedback and stable atmospheric conditions (Barnes and Polvani, 2015). In addition, reduced stratospheric water vapor enrichment (Fig. 2c–d) and the cooler Arctic surface (Hegde et al., 2025) can also contribute to the amplified Arctic cooling response."

In their reply to my comment on implementation the authors argue their "study represents a theoretical exploration using idealized numerical experiments and focuses specifically on understanding the physical mechanisms and climate response …". This seems inconsistent with several statements in the Summary and Outlook section. There the authors discuss how the injection could be done technically ("reusable rockets", "benign emissions from a hydrogen fueled rocket platform"). So it seems to me the authors try to find arguments for their proposal and, by refusing to talk about costs, ignore potential arguments against it. I'm not talking about a detailed cost study, but if "the concept is well within the scope of current technology", as the authors claim, it can't be too hard to come up with a back of the envelope estimate.

we rewrote this part:

"Based on SSP2-4.5 scenario, achieving the 1.5-degree temperature goal would require an annual SO2 injection rate of 3-8 Tg/year during 2040-2060 (Macmartin et al., 2022). Delivering 3-8 Tg of SO2 per year to 50 km altitude could, in principle, be achieved with a fleet of 30-80 reusable rockets each with a 500-ton payload, and each launched every other day. Although detailed engineering analysis of a 50 km SAI injection suborbital launch system has not yet been done, the concept is technically plausible given current and emerging spaceflight technologies (Chang and Chern, 2021; Larson et al., 2017) and recent spaceflight experience. Indeed, the requirements of a SAI50 rocket-based injection system overlap with requirements and goals of other technologies such as rapid point-to-point rocket cargo that require low-cost routine operations (Chang and Chern, 2021). Our discussion is intended to highlight the potential plausibility and physical implications of high-altitude delivery, rather than to provide an engineering design or cost assessment, which would require dedicated analyses in future work."

**#RC2**

The authors have largely addressed my concerns, but some points need further clarification.

In my first review, I had the following three main concerns:

(1) From the original manuscript, I got the impression that the H2SO4-H2O droplets proposed for SAI formed at the stratopause, even though temperatures at these altitudes are so high that the formation of H2SO4-H2O droplets is unlikely.

(2) Based on this impression, I criticized that the initial fall velocity of the particles would be very high, which poses a major complication for the model that would need to be addressed.

(3) Finally, I believed that the photolysis of H2SO4, which was neglected in this modeling, could massively alter the model results.

In the revised version, the authors address these three concerns as well as all individual comments.

Concerns (1) and (2):

The authors clarify that my impression that the H2SO4-H2O droplets in their model form at the stratopause was a misunderstanding.

The "Results" section now states: "It's important to note that while SO2 is injected at 50 km, the actual sulfate aerosol formation occurs at much lower altitudes (primarily between 10-30 km) due to the rapid transport of precursor gases and more favorable conditions for aerosol formation at lower altitudes." This is also clearly illustrated by the new Figure S3 in the supplementary information. Thank you. This is indeed in line with my expectations and does not change the potential value of the proposed new scheme. It also essentially addresses my concerns regarding points (1) and (2). However, the false impression has not been completely dispelled. The fifth sentence of the abstract states: "In SAI50, the mean meridional overturning circulation near the stratopause rapidly transports aerosols to mid-high latitudes...". This still sounds as if the circulation near the stratopause transports aerosol particles to higher latitudes, which is not the case. Rather, the circulation near the stratopause transports gaseous SO2 to higher latitudes, from where the SO2 (plus some already formed gaseous H2SO4) is transported further to lower altitudes, finally oxidized completely to H2SO4 and forming H2SO4-H2O droplets only below an altitude of 30-35 km (through bimolecular homogeneous nucleation or heterogeneous nucleation, e.g., on meteorite dust particles).

Thanks. It is revised to be:

"In SAI50, the mean meridional overturning circulation near the stratopause rapidly transports gaseous SO2 to mid-high latitudes, preventing sulfate aerosol accumulation in the tropical lower stratosphere."

The confusion continues in the Introduction, with the following statements: "To minimize the Antarctic ozone loss, it is essential that some sulfate aerosols from the intervention remain at high altitudes in the polar stratosphere. By doing so, highaltitude sulfate aerosols reduce NOx levels... In addition, aerosols formed at higher altitudes are rapidly transported to the mid-high latitudes rather than accumulating in the tropical lower stratosphere." I cannot see that aerosol particles, which form at higher altitudes, are then "rapidly transported to mid-high latitudes". I think the new Fig. S3d suggests instead that the particles are already at high latitudes when they nucleate and do not need to be transported there. This description has the potential to mislead readers and should be improved before publication.

We rewrote this sentence in Line73:

"In addition, sulfate aerosol concentrates in the mid-high latitudes rather than accumulating in the tropical lower stratosphere."

Concern (3):

In response to my concern that photolysis of H2SO4 must not be neglected as it could massively alter the results, the authors present simulations with and without H2SO4 photolysis. First, I am surprised that they can do this so easily, as I had assumed that the previous neglect was due to the model not containing H2SO4 photolysis. Since this is obviously not the case, I wonder why they did not show all results including H2SO4 photolysis right away. Second, I am even more surprised about their result, namely that H2SO4 photolysis is completely negligible. Because this contradicts my statement of a "massive" effect, I would have expected them to discuss possible reasons for this contradiction. As my statement referred to background conditions (non-SAI), we reran our CCM (SOCOL) with SAI and found that the large impact reduces to <10% in sulfate concentration in the center of the aerosol layer. This confirms the figure shown by the authors. Now, that they have demonstrated that H2SO4 photolysis indeed plays a negligible role under SAI conditions, I do not understand why the authors continue to write "Note that the photolysis of H2SO4 gas ... is not included in the model." Unless readers refer to this review, they may ask themselves the same questions I did.

Deleted.

I agree with publication in ACP, provided that the misleading statements regarding the aerosol formation are corrected and the consequences of neglecting H2SO4 photolysis are mentioned.